## [Decision Letter · Decision Letter 0]

17 Dec 2025

TRIM29 knockout pigs exhibit enhanced broad-spectrum disease resistance by amplifying type I interferon antiviral defenses

PLOS Pathogens

Dear Dr. Yang,

Thank you for submitting your manuscript to PLOS Pathogens. After careful consideration, we feel that it has merit but does not fully meet PLOS Pathogens's publication criteria as it currently stands. Therefore, we invite you to submit a revised version of the manuscript that addresses the points raised during the review process.

We look forward to receiving your revised manuscript.

Kind regards,

Wolfram Brune

Academic Editor

PLOS Pathogens

Blossom Damania

Section Editor

Editor-in-Chief

PLOS Pathogens

PLOS Pathogens

orcid.org/0000-0002-7699-2064

**Additional Editor Comments:**

All three reviewers highlighted the significance of the study and its importance for virologists and immunologists. However, they also raised a number of questions and concerns. While most of them can be addressed by changing the text or re-analyzing existing data, a few points warrant particular attention. Reviewer #3 asks for single-step and multi-step viral replication kinetics in the presence or absence of TRIM29; reviewer #2 asks that alveolar macrophages from TRIM29 KO pigs should be tested for increased susceptibility to LPS-induced inflammation (point #4). Reviewer #1 suggests that the PCR-genotyping may have to be repeated (point #1).

**Journal Requirements:**

At this stage, the following Authors/Authors require contributions: Xiaohui Yang, Jie Cheng, Huijie Jiang, Jiayong Tan, Cuizhen Wang, Changxu Song, Gengyuan Cai, Huaqiang Yang, and Zhenfang Wu. Please ensure that the full contributions of each author are acknowledged in the "Add/Edit/Remove Authors" section of our submission form.

https://journals.plos.org/plospathogens/s/submission-guidelines#loc-parts-of-a-submission

- TM on page: 21.

5) We have noticed that you have uploaded Supporting Information files, but you have not included a list of legends. Please add a full list of legends for your Supporting Information files after the references list.

Potential Copyright Issues:

i) Please confirm (a) that you are the photographer of 3E, and 4B, or (b) provide written permission from the photographer to publish the photo(s) under our CC BY 4.0 license.

7) Please amend your detailed Financial Disclosure statement. This is published with the article. It must therefore be completed in full sentences and contain the exact wording you wish to be published.

**Reviewers' Comments:**

Reviewer's Responses to Questions

**Part I - Summary**

Reviewer #1: Yang et al. investigate the role of TRIM29 in the antiviral interferon response against DNA/RNA viruses in the pig. Importantly, they generate TRIM29 KO pigs and show that these have a higher resistance towards severe/lethal infection with pseudorabies virus (PRV).

The novelty of the manuscript does not lie in the recognition of TRIM29 as an important regulator of the interferon response, since this was already known and could be expected to be also the case in pigs. The novelty mainly lies in the generation of TRIM29 KO pigs and their increased resistance to PRV infection and possibly infection by other viruses.

The data in general are of good quality, and the paper will be of interest for the broad virology and immunology community.

However, I do have some concerns, mainly related to (i) some unclear/less convincing images, (ii) unclear infection time points for some of the in vivo analyses and (iii) lack of data regarding possibly enhanced pro-inflammatory cytokine responses of TRIM KO pigs/PAM upon stimulation.

Reviewer #2: This manuscript reports an ambitious and technically impressive study integrating cellular assays, mouse models, and the successful generation and phenotypic characterization of TRIM29-knockout pigs. The authors convincingly demonstrate that TRIM29 deficiency markedly enhances type I interferon signaling and confers broad antiviral resistance, most strikingly against pseudorabies virus (PRV) in vivo, and also against RNA viruses (VSV and TGEV) in vitro. The work is timely, addresses a major challenge in swine breeding for disease resistance, and provides valuable new insights into both antiviral innate immunity and the creation of gene-edited livestock with enhanced disease resilience.

The study is well-structured, supported by extensive experimental data, and establishes clear translational links across in vitro, mouse, and porcine systems. Notable strengths include the production of cloned TRIM29-knockout pigs with rigorous off-target analysis and the execution of PRV challenge experiments under field-relevant conditions.

Overall, the manuscript has strong potential to make a significant impact in the fields of antiviral innate immunity and gene-edited disease-resistant livestock. However, several revisions are required before it can be considered for acceptance.

Reviewer #3: The central hypothesis of this paper is that TRIM29 (as previously demonstrated) is a negative regulator of IFNI and IFNIII production. After mechanistical analysis (previously demonstrated) they generated knockout animals both mice and pigs, that they then went on to challenge in vivo. The results of the animal experiments show either a delayed infection or resilience. However, due to the endpoints and the lack of multi-cycle infections this cannot properly be analysed.

This work is important but there are essential experiments needing added to this large body of work. When manipulating the interferon response, it is important to realise that this is a finely balanced system and that changing it has potentially negative effects, such as virus adaptation. This is a fact that is a well-established problem both for interferon treatment as well as cross-species transmission. I would therefore caution against a few things; the animals generated are not resistant, they are resilient and I would suggest changing this word use throughout. They still replicate virus and viral infections still have detrimental effects. Generation of a cohort of animals with such resilience is highly likely to cause virus evolution (as viruses do for example in bats where interferon expression is constitutively active at low levels).

There are some concerns about the lack of methodological information, particularly on in vitro and ex vivo experiments relating to figures 1, 2, and 7.

The authors routinely use significantly uneven group sizes and whilst correct statistical methods are mentioned in the statistical analysis section to address this issue, they don’t seem to have been used in the respective data.

In the first section they propose mechanistic analysis of TRIM29 regulation and claim novel findings of degradation of STING. However, this has already been shown in their cited reference 17. This doesn’t necessarily detract from their findings but it’s not novel. Interestingly, this paper then goes on to do a lot of animal experimentation using knockout animals, an experiment they hadn’t previously performed in vitro. This would have been the novel experiment to perform and to assess the differential response in vitro including compensatory expression since sometimes alternative pathways take over.

Abstract:

We demonstrated previously that TRIM29 functions as a negative regulator…

This finding is not novel but a previous finding and should be highlighted as such.

Introduction:

Intensive farming in the first line is a bit ambivalent. On one hand, the large number of animals needing reared globally to supply demand for animal protein has increased animal density, which leads to easier spread of infectious disease. Inversely though, biosecurity measures and stratified vaccination in more intense farming systems have reduced infectious disease burden. For example, this is very stringently observed in the spread and prevention of ASFV, which affects backyard, low intensity systems far more severely. I suggest rewording this phrase as it is misleading.

Editing of receptors: The authors allude only to complete knockouts being effective in preventing disease resistance but it is important to realise that knockouts have potential biological consequences. For CD163 knockout these may be an overreaction to allergens, e.g. dust mites, sensing and responding to staphylococcus aureus infections, and mitigation of ischaemic insults, amongst other immune response regulation. There are alternative approaches, such as manipulating individual sections for CD163 for PRRSV resistance (10.1371/journal.ppat.1006206, 10.1128/jvi.00415-18), which is also the edit that is being commercially implemented (10.1089/crispr.2023.0061), changes to individual amino acids in DNAJC14 for CSFV resistance (10.1016/j.tibtech.2025.09.008). Similar approaches to knocking out ANPEP for TGEV and PDCoV resistance (10.1007/s11248-018-0100-3) highlight the risk of delayed mammary gland development, which could impact productivity significantly. The authors also say that transgenesis is relying on random integration, which is no longer the case when using targeted approaches. This section needs updating to reflect the current state of the field and highlight positives and negatives to genome editing.

Interferon overexpression and interferon treatment: The authors cite several papers highlighting the overexpression of interferon in mice (e.g. 11) to enhance viral resilience or the interferon treatment used in hepatitis C treatment. What they don’t highlight in conjunction are the side effects of these treatments and overexpression (chronic inflammation in mice, severe side effects, including severe nausea in patients), not to also highlight viral escape from interferon treatment as discussed at length for example in reference 7.

I think it is the lack of discussing the positives and negatives of these different approaches that hampers the interpretation of results and the critical discussion of the approach taken in this paper. Enhancing the differential analysis of both genome editing, effects of interferon pathway manipulation on the host as well as the virus are critical to this paper.

Results:

Figures 1 and 2: I am going to discuss figures 1 and 2 in the same block as they are both basically conducting the same experiments, one with PRV the other with VSV. The authors conduct a series of experiments showing the induction of TRIM29 upon infection by PRV and the suppression when infected with VSV but both lead to an increase in interferon beta production though more excessive in the case of VSV. This may be due to the different mechanism of replication of these two very different viruses. The authors then performed knockdown and overexpression experiments and show increased IFNB production when TRIM29 is knocked down upon PRV and VSV infection, and decreased IFNB upon PRV and VSV infection when TRIM29 is overexpressed. Unfortunately, it’s difficult to interpret all these results since i) there are no methodological details in the M&M section on how transfections and overexpression so effects of immune stimulation through transfection methods cannot be assessed by the reviewer; ii) there’s a lack of a control, e.g. poly I:C transfection or interferon stimulation to see how knockdown of the ISG TRIM29 actually gets downregulated when it’s stimulated, i.e. actually expressed; iii) there is no virological quantification of virus produced and there’s a bit of a mixed effect studied here due to the titre used (MOI 0.1), which means there’s both an effect from bystander stimulation as well as primary infection. None of these experiments perform the knockout claimed to be successful later in mice and pigs in vitro, an easy experiment to do.

There are essential experiments that must be performed here; effect of TRIM29 overexpression and knockdown on single round infection (i.e. high MOI=4 – I would guess, very little effect), and low MOI multi-round (<=0.1) and monitor virus production over time.

Figures 3, 4, and 5 are fine.

Figure 6: I am highly surprised given the high interferon levels in the lung, to not see any signs of inflammation. This seems baffling to me also in comparison to the relatively minor differences in viral load in the lung. Please highlight at what day post challenge these images were taken and highlight what viral titres these representative animals were showing. Or, show a cross section of animals.

Figure 7: As highlighted for figure 1, this needs an analysis of multi-round infection and comparison of high vs. low MOI.

Figure 8: Normal cytokine levels don’t really add any information to the animals as standard conditions can be highly variable still. I don’t think this figure is relevant. Measuring these parameters under infection conditions would be far more informative. If the authors want to keep this in, I think it should go into supplementary information.

Materials and methods:

Please state the origin of PK-15 cells.

Severely lacking methods for figures 1 and 2, and 7.

The statements of ethical approval for all studies should be separated from the mouse section and highlight the procedure in more detail to show responsible animal experimentation.

"The animals and procedures used in this study were in accordance with the

guidelines and approval of the Institutional Animal Care and Use Committees at

South China Agricultural University." is vague and seems limited to the generation of edited mice in this context. There is no such statement for pigs and for the infection experiments.

There is no mention of humane endpoints or severity of this infection. The fact that animals died in 3-4 days in a PRV infection shows this to be quite a severe model. Heterozygosity in the infection experiment with PRV showed no difference. It is therefore questionable whether it was warranted to include a heterozygous cohort in the other infection experiments.

**Part II – Major Issues: Key Experiments Required for Acceptance**

Reviewer #1: 1. Figure 1P and also Figure 1K: what is the vertical dotted line in these Western blots? In case this would suggest that the blots were cut at these sites and, as a result, the images are composites of different blots, then I think this assay should be repeated without cutting blots -- this is particularly the case since the protein band intensities on the right side of the blot (virus-infected/TRIM29 KD) should be compared directly to the protein band intensities on the left side of the blot (especially virus-infected/vector control).

2. In addition, based on Figure 1P, the authors conclude that TRIM29 KD results in 'markedly increased protein levels of STING, phosphorylated TBK1 (pTBK1), ... in TRIM29-knockdown cells following PRV infection' (p5, no line numbers provided but second paragraph). In my opinion, the corresponding blots do not show a marked increase in STING protein or a marked increase in pTBK1. Likewise, on the right blot (TRIM29 overexpression), the blot does not seem to show the 'significantly reduced STING expression....consequently impairing TBK1-IRF3 signaling activation' (p5, also second paragraph) as claimed by the authors.

Did the author perform independent repeats of these assays and did they perform quantification and statistical analysis (especially since they mention 'significantly reduced STING expression'?

To a somewhat lesser extent, these comments also apply to Figure 2K (VSV instead of PRV).

3. For several of the analyses of the in vivo assay, it is unclear which time point of infection was analyzed and whether corresponding time points were analyzed for the different groups that are compared. For example:

- p8, bottom paragraph (and Fig 5E and 5F): 'Viral loads run the brain and lung were quantified via qPCR....'. However, both groups (WT versus TRIM29 KO pigs) must have been analyzed at different time points post infection since pigs did at different time points in the two groups (see Figure 5A and Figure 5B).

Of course, one cannot compare or make conclusions based on virus titers that were gathered at different time points of infection.

- This is also true for Fig 6C and 6D where interferon levels in brain and lung are compared between PRV-infected WT and TRIM29 KO pigs but pigs in these different groups died/were euthanized at very different time points, so this seems to make no sense.

- Also, it is unclear which time point of infection is shown in the histological images in Fig 6F and 6G and how representative these images are

4. The author rightly indicate that in mice, TRIM29 KO has been associated with increased susceptibility to e.g. LPS-induced inflammation and H. influenzae infection. However, although this could be a very important element also in pig, the authors do not (attempt to) assess this. I believe the authors should at least check whether porcine alveolar macrophages (PAM) from WT versus TRIM29 KO pigs display differences in pro-inflammatory cytokine production upon stimulation with e.g. LPS. They have collected PAMs and performed assays with these cells (Figure 7), so this seems to be a simple and very straightforward assay that would yield important information.

Reviewer #2: The authors conclude that TRIM29-knockout pigs exhibit “broad-spectrum” antiviral resistance. Currently, this claim rests primarily on resistance to a single DNA virus (PRV) in vivo and only two RNA viruses (VSV and TGEV) tested in porcine cells. To robustly support the broad-spectrum designation, particularly in the pig model, the authors should expand the ex vivo viral challenge panel (Fig. 7 and related experiments) to include additional economically important swine DNA and RNA viruses, such as African swine fever virus (ASFV) and porcine reproductive and respiratory syndrome virus (PRRSV), or at minimum provide primary cells from the knockout pigs challenged with these pathogens.

Reviewer #3: The in vitro and ex vivo experiments focus on gene expression of TRIM and type I interferons and look at an individual time point post infection (relatively early) using a low MOI infection. Furthermore, the in vitro experiments have only been performed with knockdown and overexpression experiments rather than knockout.

Interferon response manipulation as shown many times in vitro often only transiently changes virus replication, i.e. there is a delay in replication rather than an abrogation in replication. The authors must perform both high MOI single-round infections as well as multiple-round low MOI infections and monitor virus production over time in these systems. In PAMs this may be limited due to the limited lifespan of the cells but three days should be feasible.

Overall, the results are very "positively" interpreted rather than critical. Whilst mortality was reduced, there was still mortality and replication of the virus in TRIM29 knockout animals. These animals were resilient but not resistant to infection. In an infection event, i.e. upon stimulation of the ISG TRIM29, inflammation levels were elevated in knockout animals. Analysis of these factors is unlikely to show any differences when TRIM29 is not stimulated but there may or are likely differences in animals with an inflammatory or infection event. This could have negative consequences.

Further aspects for improvement have been highlighted above.

**Part III – Minor Issues: Editorial and Data Presentation Modifications**

Reviewer #1: 1. Figure 5D: all differences have the exact same p-value, which seems impossible.

2. Figure 8A shows some weird, aberrant values for some cytokines (particularly IL1beta - where 3 WT pigs seem to show at least 30x increased basal levels of IL1beta compared to the other 2). What is the authors' explanation for these values?

Reviewer #2: 1. Fig. 3A: The indicated 531 bp band does not align with the provided DNA ladder. Please repeat the genotyping PCR and provide a clearer gel image.

2. Fig. 3L: The x-axis is labeled “Hours post infection (d)”, which is confusing. Please clarify whether the time points are in hours or days and correct the label accordingly.

3. Fig. 5D: The reported p-values are identical across the 2, 4, 6, and 8 groups, which appears erroneous. Please re-calculate the statistical comparisons and update the figure.

4. Throughout Fig. 7 and its legends: “PAM” and “PAMs” are used inconsistently for porcine alveolar macrophages. Please standardize the terminology throughout the manuscript and figures.

5. Fig. 8B and 8D: Inconsistent units are used for cytokine concentrations (µg/mL in 8B vs pg/mL in 8D). Please verify and correct to consistent units (typically pg/mL for these cytokines).

6. An important study demonstrating a critical role of TRIM29 in restraining NK cell responses against the DNA virus MCMV in mice (PMID: 31270148) is not cited. The authors should discuss this work and clarify how their findings in pigs align with or differ from the reported NK phenotype in mice.

7. Materials and Methods: The section “Measurement of viral copies in tissues, serum and swab samples” describes qPCR protocols for PRV and TGEV but omits details for VSV quantification. Please add the corresponding qPCR primer/probe sequences and conditions for VSV.

Reviewer #3: Figure legends should be checked and the group sizes indicated. Furthermore, representative images should highlight the collection date and further information respective to the datapoints. Statistical methods should be considered in view of varying sample sizes and varying sample sizes highlighted in the results section as they can affect outcomes significantly.

PLOS authors have the option to publish the peer review history of their article (what does this mean? ). If published, this will include your full peer review and any attached files.

**Do you want your identity to be public for this peer review?** For information about this choice, including consent withdrawal, please see our Privacy Policy .

Reviewer #1: No

Reviewer #2: No

Reviewer #3: No

**Figure resubmission:**

**Reproducibility:**



---

## [Editor Report · Decision Letter 1]

22 Feb 2026

Dear Prof. Yang,

We are pleased to inform you that your manuscript 'TRIM29 knockout pigs exhibit enhanced broad-spectrum disease tolerance by amplifying type I interferon antiviral defenses' has been provisionally accepted for publication in PLOS Pathogens.

Best regards,

Wolfram Brune

Academic Editor

PLOS Pathogens

Blossom Damania

Section Editor

PLOS Pathogens

Sumita Bhaduri-McIntosh

Editor-in-Chief

PLOS Pathogens

orcid.org/0000-0003-2946-9497

Michael Malim

Editor-in-Chief

PLOS Pathogens

orcid.org/0000-0002-7699-2064

The word "resistant" was replaced by "resiliant" as requested by Reviewer #3. In some places, the authors replaced "resistant" by "tolerant". This is inaccurate and should be corrected.
---

## [Editor Report · Acceptance letter]

Dear Prof. Yang,

We are delighted to inform you that your manuscript, "TRIM29 knockout pigs exhibit enhanced broad-spectrum disease tolerance by amplifying type I interferon antiviral defenses," has been formally accepted for publication in PLOS Pathogens.

Best regards,

Sumita Bhaduri-McIntosh

Editor-in-Chief

PLOS Pathogens

orcid.org/0000-0003-2946-9497

Michael Malim

Editor-in-Chief

PLOS Pathogens

orcid.org/0000-0002-7699-2064